# Effects of Low-Temperature Acclimation on Nutrients of Bumble Bee *Bombus terrestris* Queens during Prediapause and Diapause

**DOI:** 10.3390/insects14040336

**Published:** 2023-03-30

**Authors:** Mengnan Shi, Kun Dong, Jie Wu, Jiaxing Huang

**Affiliations:** 1State Key Laboratory of Resource Insects of Agriculture and Rural Affairs, Institute of Apicultural Research, Chinese Academy of Agricultural Sciences, Beijing 100093, China; 2Key Laboratory for Insect-Pollinator Biology of the Ministry of Agriculture and Rural Affairs, Institute of Apicultural Research, Chinese Academy of Agricultural Sciences, Beijing 100093, China; 3College of Animal Science and Technology, Yunnan Agricultural University, Kunming 650201, China

**Keywords:** *Bombus terrestris*, low-temperature acclimation, diapause, nutrients

## Abstract

**Simple Summary:**

Bumble bees live through two phases during their life cycle: single queens and colonies. Queens enter diapause after mating as a survival strategy to avoid harsh environmental conditions. The energy requirements of queens during diapause depend mainly on the nutrient reserves at the prediapause stage. Temperature acclimation can improve the queen’s survival during diapause. Therefore, it is important to understand the effects of temperature acclimation on the prediapause nutritional reserves and energy consumption of bumble bee queens. In this study, we evaluated the effects of three acclimation temperatures and three acclimation times on the free water, protein, lipid, and total sugar reserves of bumble bee queens during prediapause and the energy consumption of queens during the 3-month diapause period. Overall, lower temperature acclimation increased queens’ lipid reserves during prediapause and reduced the consumption of protein, lipids, and total sugars during diapause. Therefore, low-temperature acclimation could be more beneficial for bumble bee queens to improve cold resistance and increase reserves of major nutrient lipids during diapause.

**Abstract:**

A queen’s diapause is a key period of the bumble bee life cycle that enables them to survive under unfavorable conditions. During diapause, queens fast, and nutritional reserves depend on the accumulation of nutrients during the prediapause period. Temperature is one of the most important factors affecting queens’ nutrient accumulation during prediapause and nutrient consumption during diapause. Here, we used a 6-day-old mated queen of the bumble bee *Bombus terrestris* to evaluate the effect of temperature (10, 15, and 25 °C) and time (3, 6, and 9 days) on free water, protein, lipids, and total sugars during prediapause and at the end of 3 months of diapause. Stepwise regression analysis revealed that total sugars, free water, and lipids were much more affected by temperature than protein (*p* < 0.05). Lower temperature acclimation significantly increased (*p* < 0.05) free water and lipid accumulation by queens during prediapause. In contrast, higher temperature acclimation significantly increased (*p* < 0.05) protein and total sugar accumulation by queens during prediapause. The effect of temperature acclimation on the queen survival rate was not significantly different (*p* > 0.05) after 3 months of diapause. Moreover, lower temperature acclimation reduced protein, lipid, and total sugar consumption by queens during diapause. In conclusion, low-temperature acclimation increases queens’ lipid accumulation during prediapause and reduces the nutritional consumption of queens during diapause. Low-temperature acclimation during prediapause could benefit queens by improving cold resistance and increasing reserves of major nutrient lipids during diapause.

## 1. Introduction

Bumble bees are pollinators that play important roles in agricultural and natural ecosystems [1]. They are social insects, but at the end of the colony’s life cycle, the next generation of queens depart from the nest, mate with drones, and enter diapause [2]. Bumble bee queens’ diapause is an adaptive mechanism to environmental changes. It may also be an inherent genetic trait of queens [2] that allows them to synchronize their period of activity with a suitable environment [3,4,5]. Prediapause is the critical period for the queen’s nutrition accumulation. The amount or type of nutrition the queen accumulates during this period directly affects their ability to survive during diapause [6,7,8]. Temperature may be the most important factor influencing nutrient accumulation or energy acclimation at prediapause and nutrient consumption during diapause in bumble bee queens.

The effect of temperature on the prediapause period of queens is poorly understood, and only a few relevant studies have been performed previously [9,10]. The bumble bee species initially studied were susceptible to low-temperature environments [9]. Because the queens were directly transferred from a warm environment to a colder environment, it had an adverse lethal or sublethal effect on them, reducing their adaptability to environmental changes and strongly reducing their ability to survive during the diapause period [11]. Higher temperatures also lead to additional depletion of the queen’s prediapause nutrient reserves, which adversely affects the accumulation of nutrients [8,12].

Low-temperature acclimation plays an essential role in the temperature tolerance of insects [13]. Cold tolerance is usually a plastic response in insects, and exposure to relatively mild sublethal temperatures for acclimation prior to entering diapause could enhance their cold resistance [14]. In addition, insects can increase the production and accumulation of freeze-resistant substances, thus improving their survival rate during diapause [15,16]. Previous studies have shown that different temperature acclimation conditions improved the survival rate of bumble bee queens during the diapause period [17]. However, the influence of changes in nutrients after temperature acclimation and the nutritional metabolism of queens during the diapause period have not been evaluated in detail.

The aim of this study was to investigate the effect of temperature acclimation on the nutrition or energy acclimation of bumble bee queens during prediapause and nutrient consumption during diapause. This will help us to better understand the effect of temperature acclimation on the cold resistance of queens and their ability to survive during diapause. We selected three temperatures (10, 15, and 25 °C) and three time points (3, 6, and 9 days) to evaluate the effects of temperature acclimation on free water, protein, lipids, and total sugars of bumble bee *B. terrestris* queens during prediapause and at the end of 3 months of diapause. We hypothesized that bumble bee queens’ nutrient accumulation or energy acclimation during prediapause would adapt to the environmental temperature. Therefore, low-temperature acclimation would increase the accumulation of key cold-resistant substances in queens, improve their ability to survive during diapause, and reduce the nutrient consumption of queens during diapause.

## 2. Materials and Methods

### 2.1. Bumble Bee Queen Rearing

Bumble bee *B. terrestris* queens were collected from the rearing room at the Institute of Apicultural Research, Chinese Academy of Agricultural Sciences (Beijing 100089, China). We randomly selected 60 bumble bee colonies reared in wooden rearing boxes at the peak of colony development. We selected queens that emerged within 24 h (as determined by their distinct silvery appearance). Fifty virgin queens were reared in plastic boxes (27 × 16 × 15 cm) and kept in complete darkness at a temperature of 25 °C and relative humidity of 55% ± 5%. They were free to feed at any time (50% sucrose solution and fresh frozen pollen mixture). Mating was performed in flight cages (95 × 95 × 95 cm) when virgin queens were 6 days old, and drones were 10–15 days old. Mated queens were used for later treatment.

### 2.2. Bumble Bee Queen Treatment and Collection

We randomly selected 460 mated queens for the following experimental treatments:

(1) For the temperature acclimation treatment, 310 queens were randomly divided into 9 groups of 33–35 queens each. They were kept at 10, 15, or 25 °C for 3, 6, or 9 days. During the rearing period, they were free to feed (50% sucrose solution and fresh frozen pollen mixture). The surviving queens at the end of the treatment were immediately stored at −80 °C.

(2) For the diapause stage experiment, bumble bee queens of artificial rearing were put into the diapause environment on the 6th day after mating [5,18,19]. Therefore, we divided 150 queens randomly into 3 groups of 50 queens each. They were kept at 10, 15, or 25 °C for 6 days and free to feed (50% sucrose solution and fresh frozen pollen mixture). Then, each queen was individually placed in a small plastic tube (4 × 4 × 7 cm) and held in a refrigerator at a temperature of 4 °C ± 0.5 °C for 3 months. Queen survival was recorded at the end of the diapause stage, and then they were immediately stored at −80 °C.

### 2.3. Free Water Content

To determine the effect of temperature acclimation on the free water content of queens, we collected 165 that were alive at the end of the above treatments. From the temperature acclimation stage experiment, we obtained 135 queens, 15 from each treatment group. From the diapause stage experiment, we obtained 30 queens, 10 from each treatment group. According to a previous study [20], the wet weight of each queen was determined using an electronic balance accurate to 0.001 g (WW). The queens were baked to a consistent weight in a constant temperature drying oven (Tianjin Teste Instruments Co., Ltd., 101−2AB, Tianjin, China) at 60 °C for 72 h. Each queen was weighed again, and the dry weight was recorded (DW). The free water content of the queen bee was determined using the formula (WW − DW)/WW × 100%.

### 2.4. Fat Body Nutrient Content Determination

To determine the effect of temperature acclimation on the fat body nutrients (protein, lipids, and total sugars) of queens, we collected 198 that were alive at the end of the above treatments. From the temperature acclimation stage experiment, 162 queens were obtained, 18 from each treatment group, and three were considered as one sample. From the diapause stage experiment, 36 queens were obtained, 12 from each treatment group, and two queens were considered one sample. The fat body tissue of the queen bee was scraped using the queen dissection procedure as follows: the queen bee samples were thawed at room temperature, the wings and limbs were removed, and the thorax was placed in a dissecting wax tray with a needle. Using surgical scissors, the abdomen was sliced from the bottom to the top, and the abdominal carapace was stretched out to the sides and secured with pins. The gut, ovaries, and membrane material were removed from the queen bee’s abdomen. The abdomen was observed using a dissecting microscope DP Controller (Olympus, Olympus Corporation, Tokyo, Japan), and the fat body was identified as the flocculent tissue just above the carapace. The fat body was scraped clean with forceps, placed in a 2 mL centrifuge tube, immediately placed in liquid nitrogen, and stored at −80 °C for backup after all collections were completed.

### 2.5. Protein Concentration Determination

Tissue samples were removed from −80 °C, 1.5 mL of PBS buffer was added to the fat body samples, and the tissues were ground using a high-throughput tissue grinder (70 Hz, 60 s, two times). The tissue was centrifuged at 4 °C for 5 min at 600× *g* using a high-speed refrigerated centrifuge (Centrifuge 5804R, Eppendorf, Germany). Fifty microliters of tissue supernatant were taken after centrifugation and diluted with PBS buffer at appropriate volumes. Protein concentration was determined using a BCA protein quantification kit (Lab CatB5000). The protein standard solution was diluted with PBS buffer at concentrations of 25 μg/mL, 125 μg/mL, 250 μg/mL, 500 μg/mL, 750 μg/mL, and 1000 μg/mL. The horizontal coordinate is the concentration of the standard, and the vertical coordinate is the absorbance value (OD) to plot the protein standard curve. BCA Reagent A (BCA disodium salt, 2% anhydrous sodium carbonate, 0.16% sodium tartrate, 0.4% sodium hydroxide, 0.95% sodium bicarbonate, mixed to pH 11.25) and BCA Reagent B (4% copper sulfate) from the kit were used to prepare the protein working solution in the ratio of 50:1. Next, 200 μL of protein working solution was added to 20 μL of sample solution, and the color development reaction was performed for 30 min in a 37 °C thermostat. The OD was measured at 595 nm using an enzyme standard meter (SoftMax pro i3, Molecular Devices, San Jose, CA, USA), and the protein concentration was calculated from the protein standard curve.

### 2.6. Lipid Concentration Determination

The concentration of lipids in fat bodies was determined according to the method previously described [21]. After centrifugation, 100 μL of the tissue supernatant was taken and diluted to appropriate volumes. Then, 180 μL of the diluted solution was placed into a 2 mL centrifuge tube, 20 μL of 20% sodium sulfate solution was added, and 1.5 mL of the chloroform–methanol mixture (chloroform:methanol = 1:2) was added. The samples were shaken vigorously for 1 min and centrifuged at 4 °C and 600× *g* for 15 min. Then, after centrifugation, 100 μL of the tissue supernatant was placed into a 1.5 mL centrifuge tube and heated in a metal bath (Hangzhou You Ning Instruments Co., Ltd., GL100, Hangzhou, China) at 90 °C for 15–20 min until complete evaporation. Then, 20 μL of 98% concentrated sulfuric acid solution was added and heated in a constant temperature water bath (Shanghai Sen shin Experimental Instruments Co., Ltd., DK-8D, Shanghai, China) at 90 °C for 2 min. The solution was then removed and cooled on ice, 180 μL vanillin solution was added, and it was incubated for 15 min at room temperature. Cholesterol standards (Shanghai Yuan Ye Biotechnology Co., Ltd., Shanghai, China) were diluted with a chloroform–methanol mixture (chloroform:methanol = 1:2) at concentrations of 0 μg/mL, 200 μg/mL, 400 μg/mL, 600 μg/mL, 800 μg/mL, and 1000 μg/mL. The horizontal coordinate is the concentration of the standard, and the vertical coordinate is the OD to plot the lipid standard curve. The OD values were measured at 525 nm using an enzyme marker, and the lipid concentrations were calculated from the lipid standard curve.

### 2.7. Total Sugar Concentration Determination

The concentration of total sugars in the fat bodies was determined according to the method previously described [22,23]. After centrifugation, 20 μL of tissue supernatant was added to a 96-microwell quartz plate, 230 μL of anthrone reagents (ready-to-use) was added, the plate was sealed with a sealing film, and then incubated for 15 min at room temperature and 15 min at 90 °C in a constant temperature water bath. The glucose standard (Sigma-Aldrich, Saint Louis, MO, USA) was diluted to 0 μg/mL, 200 μg/mL, 400 μg/mL, 600 μg/mL, 800 μg/mL, and 1000 μg/mL. The horizontal coordinate is the concentration of the standard, the vertical coordinate is the OD, and the glucose standard curve was plotted. The OD value was measured at 625 nm using an enzyme marker, and the total sugar concentration was calculated from the glucose standard curve.

### 2.8. Data Analysis

The statistical analyses were performed in SPSS 26.0 software, and only *p* < 0.05 was considered significant. All the results were visualized using GraphPad Prism 9.4.1. We used the Shapiro–Wilk test to determine the normality distribution of the data, and all data were tested to show a normal distribution. The effects of temperature and time on the investigated substances of queens were tested by general linear model analysis. The effects of temperature acclimation on the survival of diapause queens were tested by chi-square analysis. Changes in the investigated substances of queens after acclimation and at the end of diapause were compared by independent samples t-tests to determine the effect of temperature acclimation on the consumption of the investigated substances during diapause. The degree of the effects of temperature and time on the investigated substances of queens during prediapause were analyzed by stepwise regression models. We removed outliers (data points outside the 95% confidence interval) from the data analysis prior to all analyses. The data are presented as the mean ± standard deviation (mean ± SD).

## 3. Results

### 3.1. Effect of Low-Temperature Acclimation on the Diapause Survival Rate

Temperature acclimation for 6 days during prediapause had no significant effect on the survival rate of queens after 3 months of diapause (x^2^ = 3.433, df = 2, *p* > 0.05). The measured queen survival rate was 98% at 15 °C acclimation, followed by 96% at 10 °C and 90% at 25 °C acclimation.

### 3.2. Effect of Low-Temperature Acclimation on Free Water Content

Temperature and time had interactive effects on the free water content of prediapause queens (F = 5.19, df = 4, *p* < 0.01). Temperature had a significant effect on the free water content of queens (F = 245.142, df = 2, *p* < 0.001), while time had no significant effect (F = 2.620, df = 2, *p* > 0.05). Low-temperature acclimation significantly decreased (*p* < 0.05) the free water content of queens during prediapause. In comparison to 25 °C acclimation, the free water content of queens increased by 8%, 9%, and 8% at 10 °C acclimation for 3, 6, and 9 days, respectively, and increased by 4%, 3%, and 1% at 15 °C acclimation for 3, 6, and 9 days, respectively (Figure 1A). Temperature acclimation for 6 days during prediapause influenced the change in free water of queens during diapause (15 °C: t = −1.092, *p* < 0.05; 25 °C: t = −8.402, *p* < 0.001). The free water content increased by 7% and 8% at 15 °C and 25 °C acclimation, respectively (Figure 1B).

### 3.3. Effect of Low-Temperature Acclimation on Protein Concentration Level

Temperature and time had no interactive effect on protein concentration in the fat bodies of prediapause queens (F = 1.010, df = 4, *p* > 0.05). However, temperature had a significant effect on the protein concentration in the fat bodies (F = 49.540, df = 2, *p* < 0.001), and time also had a significant effect (F = 24.578, df = 2, *p* < 0.001). High-temperature acclimation significantly increased (*p* < 0.05) the protein concentration of queens during prediapause. In comparison to 10 °C acclimation, the protein concentration of queens increased by 15%, 18%, and 21% at 15 °C acclimation for 3, 6, and 9 days, respectively, and increased by 7%, 14%, and 17% at 25 °C acclimation for 3, 6, and 9 days, respectively (Figure 2A). Temperature acclimation for 6 days during prediapause influenced the protein consumption of queens during diapause (10 °C: t = 2.965, *p* < 0.05; 15 °C: t = 2.377, *p* < 0.05; 25 °C: t = 8.439, *p* < 0.001). The protein consumption was 10% at 10 °C, 2% at 15 °C, and 22% at 25 °C. Compared to 25 °C acclimation, the protein consumption was reduced by 12% and 20% at 10 °C and 15 °C acclimation, respectively (Figure 2B).

### 3.4. Effect of Low-Temperature Acclimation on Lipid Concentration Level

Temperature and time had no interactive effect on lipid concentration in the fat bodies of prediapause queens (F = 0.182, df = 4, *p* > 0.05). However, temperature had a significant effect on the lipid concentration in the fat bodies (F = 49.070, df = 2, *p* < 0.001), and time also had a significant effect (F = 23.271, df = 2, *p* < 0.001). Low-temperature acclimation significantly increased (*p* < 0.05) lipid concentration of queens during prediapause. In comparison to 25 °C acclimation, the lipid concentration of queens increased by 29%, 34%, and 33% at 10 °C acclimation for 3, 6, and 9 days, respectively, and increased by 11%, 18%, and 18% at 15 °C acclimation for 3, 6, and 9 days, respectively (Figure 3A). Temperature acclimation for 6 days during prediapause influenced the lipid consumption of queens during diapause (10 °C: t = 10.99, *p* < 0.001; 15 °C: t = 3.839, *p* < 0.01; 25 °C: t = 10.082, *p* < 0.001). The lipid consumption of queens was 37% at 10 °C, 23% at 15 °C, and 48% at 25 °C. Compared to 25 °C acclimation, lipid consumption was reduced by 11% and 25% at 10 °C and 15 °C acclimation, respectively (Figure 3B).

### 3.5. Effect of Low-Temperature Acclimation on Total Sugar Concentration

Temperature and time had an interactive effect on the total sugar concentration in the fat bodies of prediapause queens (F = 6.430, df = 4, *p* < 0.001). Temperature had a significant effect on the total sugar concentration in the fat bodies (F = 183.633, df = 2, *p* < 0.001), and time also had a significant effect (F = 19.175, df = 2, *p* < 0.001). High-temperature acclimation significantly increased (*p* < 0.05) the total sugar concentration of queens during prediapause. In comparison to 25 °C acclimation, the total sugar concentration of queens decreased by 29%, 26%, and 27% at 15 °C acclimation for 3, 6, and 9 days, respectively, and decreased by 62%, 46%, and 33% at 10 °C acclimation for 3, 6, and 9 days, respectively (Figure 4A). Temperature acclimation for 6 days during prediapause influenced the total sugar consumption of queens during diapause (10 °C: t = 2.965, *p* < 0.05; 15 °C: t = 2.377, *p* < 0.05; 25 °C: t = 8.439, *p* < 0.001). The total sugar consumption was 46% at 10 °C, 37% at 15 °C, and 47% at 25 °C. Compared to 25 °C acclimation, the total sugar consumption was reduced by 1% and 10% at 10 °C and 15 °C acclimation, respectively (Figure 4B).

### 3.6. Effect of Low-Temperature Acclimation on Response Variables

Stepwise regression analysis showed the effect of different factors (temperature and time) on the response variables (free water, protein, lipid, and total sugar content). The effect of temperature on free water, lipid, and total sugar content was more significant (*p* < 0.05) than that of time. Temperature had the most significant effect on total sugars, followed by free water and lipids. The effects of time on protein content were more significant (*p* < 0.05) than those of temperature. However, time had no significant (*p* > 0.05) effect on free water content (Table 1).

## 4. Discussion

Temperature is a key factor influencing queens’ nutrient or energy acclimation during prediapause and nutrient consumption during diapause. Therefore, increasing the resistance of queens to cold and their ability to survive the diapause period was essential for establishing and developing bumble bee colonies. In this study, we found that temperature acclimation had a significant effect on nutrient accumulation during prediapause and nutrient consumption during diapause. Low-temperature acclimation improved the cold resistance of queens, increased their accumulation of lipids (the main energy reserve during diapause), and reduced their consumption of protein, lipids, and total sugars during diapause. To the best of our knowledge, this was the first systematic study on the low-temperature acclimation of *B. terrestris* queens.

In this study, temperature acclimation did not have a significant effect on the survival rate of queens after 3 months of diapause. Still, after 6 days of acclimation at 10 °C and 15 °C, the survival rate of queens during diapause increased by 6% and 8%, respectively, consistent with previous studies in *B. ignitus* queens [17]. Although not statistically significant, these results have some reference value for artificial bumble bee-rearing techniques.

The free water content of queens increased with decreasing acclimation temperature, which was the opposite of the response in other insects. The low temperature usually decreases the free water content of insects [24], which results from the excretion of free water or its conversion to bound water, thus increasing the body fluid concentration [25]. The free water content of queens remained at 56% during diapause, consistent with previous study results on other bumble bee species [2]. Low-temperature acclimation met the water content requirement of queens during diapause and decreased the additional conversion of lipids due to dehydration [26].

The functions of proteins, lipids, carbohydrates, and other energetic substances associated with cold resistance in insects can only be activated when there are significant changes in the external environment (e.g., temperature) to help them survive adverse conditions [27,28,29]. Previous studies showed that the protein levels of queens were maintained at the same level during the diapause process and increased significantly only during reproduction [1,2,23,30]. We found that acclimation to higher temperatures increased the protein accumulation of queens, while acclimation to lower temperatures decreased the protein accumulation. The same phenomenon was observed in other insects [7,8], with a certain range of temperature acclimation allowing queens to adapt, while out-of-range temperature acclimation leads to passive acclimation. After low-temperature acclimation, the consumption of proteins by queens was reduced during the diapause period. This was caused by increasing the antifreeze substance during the acclimation process and decreasing the conversion of protein during diapause [7,31,32].

Lipids gradually increased with decreasing acclimation temperature, which confirmed that lipids were the main energy reserve of queens during the diapause period [6,23,31,33]. The accumulation of lipids by queens occurs prior to diapause [6]. We found that lipids gradually decreased with increasing acclimation time. This was consistent with the result of preferential consumption of lipids, as confirmed in studies of other bumble bee species [2]. Lipids were converted to unsaturated fatty acids to keep the food liquid [34]. It is also converted in large amounts to glycerol to prevent tissue frostbite [35]. After low-temperature acclimation, the consumption of lipids was reduced by queens during the diapause period. The nutrients accumulated during acclimation meet the queen’s cold-resistant properties and reduce lipid conversion during diapause [7,31].

The effect of low-temperature acclimation on the total sugar accumulation of queens was the opposite of its effect on lipid accumulation. The total sugar accumulation of queens decreased with decreasing temperature acclimation, consistent with previous findings [36]. The total sugar accumulation of queens increased with time during acclimation at 10 °C. It has also been found that low-temperature acclimation induced rapidly accumulated total sugars to prepare for the nutritional requirements of diapause in other insects [27]. After low-temperature acclimation, queens’ total sugar consumption during the diapause period was reduced. In cold environments, the queens converted total sugars into protective substances such as glycerol, sorbitol, glycosylic acid, and alginate. This decreased the need for queens to convert total sugars during diapause [32,37,38].

This study confirms our hypothesis that nutrient accumulation or energy acclimation of queens during the prediapause period could be adaptively regulated according to the environmental temperature. Lower temperature acclimation improved the cold resistance of queens, increased the accumulation of lipids, and reduced nutrient consumption during diapause. Although we found a close relationship between low-temperature acclimation and nutrient reserve and the metabolism of queens, the detailed regulatory mechanisms require further investigation.

## 5. Conclusions

Temperature acclimation has an overall effect on the free water, protein, lipid, and total sugar reserves of queens during prediapause and on energy consumption during diapause. The free water, lipid, and total sugar reserves of queens during prediapause were significantly affected by temperature compared to proteins. Furthermore, higher temperature acclimation increased the protein and total sugar reserves of queens during prediapause. Moreover, lower temperature acclimation increased the reserve of lipids, the main energy reserve of queens, and the free water content during prediapause and reduced the nutrient consumption (proteins, lipids, and total sugars) of queens during diapause. In conclusion, low-temperature acclimation during prediapause could benefit queens by improving cold resistance and increasing reserves of major nutrient lipids during diapause.

## Figures and Tables

**Figure 1 insects-14-00336-f001:**
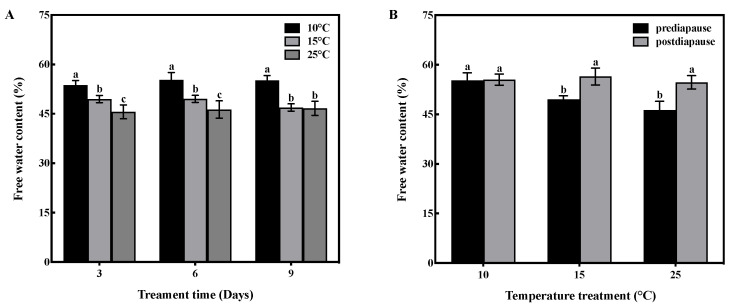
Effect of temperature acclimation on the free water content of queens. The value represents the percentage of the queen’s body weight for each sample. (**A**) Effect of temperature acclimation on the free water content of queens during prediapause. Different letters represent statistical significance between the different temperatures (*p* < 0.05). (**B**) Effect of temperature acclimation for 6 days during prediapause on changes in the free water content of queens during diapause. Different letters represent statistical significance between the different periods of diapause (*p* < 0.05).

**Figure 2 insects-14-00336-f002:**
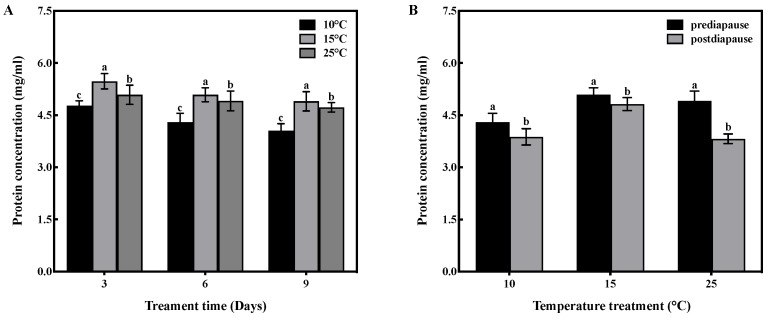
Effect of temperature acclimation on protein concentration in the fat bodies of queens. The values represent the concentrations (in mg/mL) of tissue samples of medium volume for each sample. (**A**) Effect of temperature acclimation on protein concentration in the fat bodies of queens during prediapause. Different letters represent statistical significance between the different temperatures (*p* < 0.05). (**B**) Effect of temperature acclimation for 6 days during prediapause on protein consumption of queens during diapause. Different letters represent statistical significance between the different periods of diapause (*p* < 0.05).

**Figure 3 insects-14-00336-f003:**
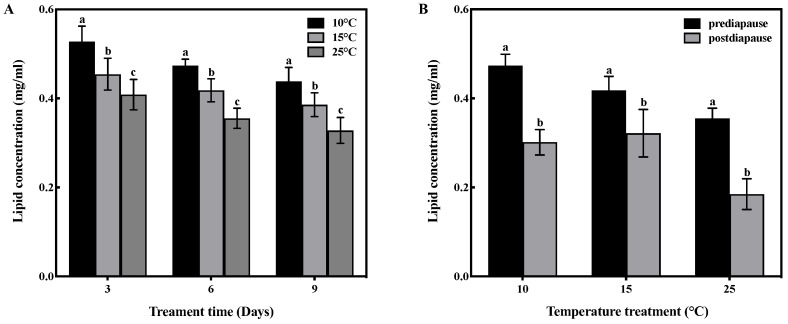
Effect of temperature acclimation on lipid concentration in the fat bodies of queens. The values represent the concentrations (in mg/mL) of tissue samples of medium volume for each sample. (**A**) Effect of temperature acclimation on lipid concentration in the fat bodies of queens during prediapause. Different letters represent statistical significance between the different temperatures (*p* < 0.05). (**B**) Effect of temperature acclimation for 6 days during prediapause on lipid consumption in the fat bodies of queens during diapause. Different letters represent statistical significance between the different periods of diapause (*p* < 0.05).

**Figure 4 insects-14-00336-f004:**
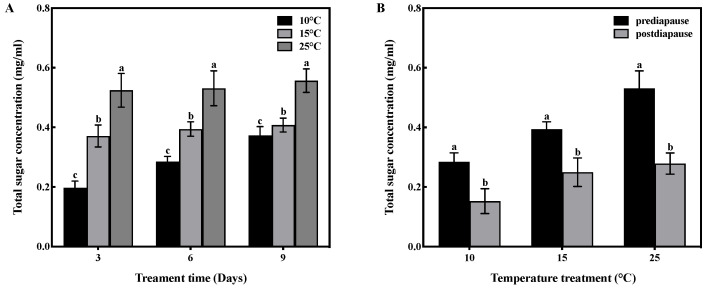
Effect of temperature acclimation on total sugar concentration in the fat bodies of queens. The values represent the concentrations (in mg/mL) of tissue samples of medium volume for each sample. (**A**) Effect of temperature acclimation on total sugar concentration in the fat bodies of queens during prediapause. Different letters represent statistical significance between the different temperatures (*p* < 0.05). (**B**) Effect of temperature acclimation for 6 days during prediapause on total sugar consumption of queens during diapause. Different letters represent statistical significance between the different periods of diapause (*p* < 0.05).

**Table 1 insects-14-00336-t001:** Stepwise regression model on the effects of temperature acclimation of free water and nutrient accumulation of queens during prediapause.

Indicators	Factors	Unstandardized Coefficients		Standardized Coefficients	*p*	R Square	Adjusted R Square	F
		B	Std Error	Bate				
**Free water**	**Time**	-	-	0.014	0.757	-	-	-
	**Temperature**	−4.276	0.228	−0.86	0.000 ***	0.74	0.737	352.021
**Protein**	**Time**	−0.092	0.019	−0.494	0.000 ***	0.244	0.229	16.764
	**Temperature**	0.266	0.057	0.473	0.000 ***	0.468	0.447	22.409
**Lipid**	**Time**	−0.013	0.002	−0.491	0.000 ***	0.756	0.747	55.241
	**Temperature**	−0.058	0.006	−0.718	0.000 ***	0.515	0.506	79.145
**Total sugar**	**Time**	0.014	0.003	0.284	0.000 ***	0.845	0.839	138.985
	**Temperature**	0.126	0.008	0.874	0.000 ***	0.765	0.76	168.812

Note: “-” indicates that the factor was excluded from the analysis. “***” indicates that there was a significant effect of 1%.

## Data Availability

The data presented in this study are available on request from the corresponding author.

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
