# Peer review of "Effects of Low-Temperature Acclimation on Nutrients of Bumble Bee Bombus terrestris Queens during Prediapause and Diapause"

_insects, 2023, doi:10.3390/insects14040336_

Round 1

Reviewer 1 Report

Low Temperature Acclimation Benefits Bumble Bee Bombus terrestris Queens During Diapause

Mengnan Shi, Kun Dong, Jie Wu, Jiaxing Huang

insects-2220256-peer-review-v1

 This is a well done and interesting paper on energy reserves in the fat body of bumblebees.

Overall it is well written, though the discussion can stand some condensing and more suitable phrasing.

I am not happy with the use of the term „cold-resistant substance“ in the paper. Ecophysiologists usually think on glycerol, polyols, sugars etc. in this context.

MATERIALS AND METHODS

L106: space missing at „6 or“

L122: delete „and documented“

L155: specify what is „protein working solution“

L199-200: here and elswhere in the paper, ‘writing physiological active substances‘ seems not appropriate because you only analysed a few of them. Therefore writing „ … on the investigated substances of queens …“ seems more appropriate.

How many (and which) outliers did you exclude from the dataset?

RESULTS

What was the weight of the fat body (samples)?

L207-209: since there was NO significant effect of acclimation on survival rate (!) it seems more appropriate to change this sentence to:
„The measured queen survival rate was 98% at 15°C acclimation, followed by 96% at 10°C and 90% at 25°C acclimation.“

L262: replace dot by colon: „(10°C:“

L282-285: please keep ‚direction‘ of comparison as used elswhere (helps the reader a lot!):
„In comparison to 25 °C acclimation, the total sugar concentration decreased by …“

DISCUSSION

The discussion can stand some editing and condensing.

It does not sound well (too unsure) if you too often write „It could be that …“, e.g. in
L372-375: write „In XXXXX it was observed that … „ or so

L379, L389: I am not happy with the use of the term „cold-resistant substance“ in the paper. Ecophysiologists usually think on glycerol, polyols, sugars etc. in this context.
L379: Better write „…
(the main energy reserve during … „
L389-390: Better write „… the main energy reserve of queens … „

Reviewer 2 Report

In this manuscript, the authors investigated the effect of acclimation temperature (10, 15 and 25°C) and times (3, 6 and 9 days) on nutrient accumulation (i.e. free water, protein, lipids, and total sugar) of Bombus terrestris queens during prediapause and on energy consumption during diapause. Overall, this work is nicely presented. However, I have a major comment for the authors to consider.

Major:

Authors stated that temperature acclimation for 6 days during prediapause had no significant effect on the survival rate of queens during diapause (lines 206-207, 214-325). It implied that the reason for selecting this acclimation time (i.e. 6 days) is insufficient for further study (i.e. Effect of temperature acclimation for 6 days during prediapause on nutrient consumption during diapause). Please state the reason for selecting this acclimation time in text (Section 2.2). Also, according to this result, the conclusion of this paper (Low temperature acclimation could be more beneficial for bumble bee queens during diapause) cannot be obtained.

Minor:

1. Lines 21-23, delect the sentence According to ...during diapause’’.

2. Line 55, delect words to survive unfavorable conditions’’. 

3. Lines 160: The sentenceThe concentration of lipids in fat bodies was determined using the  method  [24,25]” should be The concentration of lipids in fat bodies was determined according to the method previously described [24,25]”.

4. Lines 179-180: The sentenceThe concentration of total sugars in the fat bodies was determined using the method of [26,27].should be The concentration of total sugars in the fat bodies was determined using the method previously described [26,27]

5. Table1: Replace Age with Time

Reviewer 3 Report

Even bumblebee queen diapause is quite well studied, I agree with you that larger study on the low temperature acclimation of B. terrestris queens has been missing.

For better clarity, I suggest to make new paragraphs at lines 104 and 109 (starting with numbers 1 and 2). The abbreviation WW at line 122 stands for "fresh weight"? (it probably originate from "wet weight" abb.). As for the graphical display of the results, it took me a while to understand what the note "different letters represent statistical significance" means, because Fig. from 2 to 4 displays only statistically significant differences (although it is of course correct from a formal point of view). 

The article contains a clear message that can be used in commercial bumblebee production, so I have no any further major comments.

Round 2

Round 3

Reviewer 2 Report

All  questions have been reviewed. It is ok now.